# A New Method of Predicting the Parameters of the Rotational Friction Welding Process Based on the Determination of the Frictional Heat Transfer in Ti Grade 2/AA 5005 Joints

**DOI:** 10.3390/ma16134787

**Published:** 2023-07-03

**Authors:** Piotr Lacki, Janina Adamus, Wojciech Więckowski, Maciej Motyka

**Affiliations:** 1Faculty of Civil Engineering, Częstochowa University of Technology, J.H. Dąbrowskiego 69 Str., 42-201 Częstochowa, Poland; janina.adamus@gmail.com; 2Faculty of Mechanical Engineering and Computer Science, Częstochowa University of Technology, J.H. Dąbrowskiego 69 Str., 42-201 Częstochowa, Poland; wojciech.wieckowski@pcz.pl; 3Faculty of Mechanical Engineering and Aeronautics, Rzeszow University of Technology, Powstancow Warszawy 12 Av., 35-959 Rzeszow, Poland; motyka@prz.edu.pl

**Keywords:** solid-state welding, RFW, intermetallic compounds, numerical simulation

## Abstract

The article presents an original approach to determining the basic parameters of rotational friction welding (RFW) based on the analysis of friction heat transfer at the faying surfaces. Dissimilar Ti Grade 2/AA 5005 joints were used to demonstrate the method. The work established that for the analyzed joint, the optimum temperature at the faying surface that allow for a good quality weld to be obtained should be ~505 °C. On this basis, a map of optimal parameters was developed to achieve this temperature. This approach could potentially allow for more precise control of the welding process, leading to better joint quality and performance. The paper includes both a description of the technological process of friction welding and an attempt to explain the mechanism of the phenomena occurring in the welding area. The numerical calculations presented in the article were carried out using the ADINA System v. 9.8.2, which allows for the consideration of heat friction in the axial symmetric thermo-mechanical model. Frictional resistance was determined by the temperature-dependent friction coefficient. The assumed thermo-mechanical model required the determination of elastic-plastic properties versus temperature for the analyzed materials. The simulations of the friction welding were carried out for the different welding parameters and time. The different variants of friction welding were modelled.

## 1. Introduction

Rotary Friction Welding (RFW) is now a widespread welding technique, as it enables a wide range of ferrous and non-ferrous materials to be joined, such as aluminum alloys or titanium materials, to create lightweight structures [1,2,3]. Strong, lightweight, and relatively cheap structures are important for the automotive, nuclear power, aerospace, and infrastructure industries, among many other. In conventional joining techniques, metals melt, and various types of defects are created during solidification. The RFW technique belongs to solid-state welding and allows the production of strong and high-quality welds at a temperature below the melting points of the joined materials, both in case of similar and dissimilar joints [4]. RFW uses the phenomenon of friction between the welded components.

In the RFW process, heat is generated by the conversion of mechanical energy into thermal energy in the contact area of the rotating and pressed components. The weld is formed by the rapid heating of the materials due to frictional heat and upsetting in the heat-affected zone (HAZ). When the shear strength is reached, the plasticized material at the contact area of the joined parts begins to flow in a radial direction under the influence of the axial pressure and forms characteristic flash. The flash around the weld can be removed by machining or grinding. Using this technology, solid and tubular parts can be joined effectively. It is also possible to join other shapes, provided that one of the components has rotational symmetry.

### Types of Welded Materials

RFW can be used to join a wide range of materials, including metals, plastics, and composites. Dissimilar components may require special considerations and may be more challenging to weld due to their different melting points and other properties. The type of welded materials determines the process conditions, as shown in Table 1. It is easy to see that the higher the rotational speed, the shorter the time of the friction phase.

Welding aluminum and its alloys with titanium or its alloys is hindered by the significant differences in the physical and mechanical properties of these materials, especially their melting points and thermal conductivity. For these reasons, it is difficult to generate sufficient heat and plastic deformation to produce a high-quality weld. In addition, these materials differ in chemical composition and metallurgical properties, leading to formation of new phases such as intermetallic compounds (IMCs) and significantly affecting the microstructure and mechanical properties of the welded joint.

## 2. Welded Materials

In this work, an RFW joint between commercially pure (CP) titanium Grade 2 and aluminum alloy AA 5005 was selected for the analysis.

### 2.1. Characteristics of Aluminum Alloy AA 5005

The AA 5005 belongs to the aluminum alloys of the 5000 series, in which magnesium is the main alloying element (Table 2). Due to its relatively good strength and good formability, especially in soft states; good fatigue resistance; as well as good weldability and corrosion resistance, it is used as a construction material.

AA 5005-H32 Grade is strain-hardened and then stabilized to a strength that is roughly a quarter of the value between annealed (O) and fully hard (H38). It has moderate strength and good ductility, with a yield tensile strength (YTS) of 117 MPa, an ultimate tensile strength (UTS) of 138 MPa, and an elongation at break (EL) of 11% (Table 3).

The AA 5005 aluminum alloy has a low melting point of 632–654 °C, a coefficient of linear thermal expansion (CTE) of 25.6 × 10^−6^ 1/K in the temperature range 20–300 °C and a good thermal conductivity (Table 4).

The stress-strain curves at different forming temperatures for AA 5005-H3 alloy was adopted on the basis of the experiments performed by the authors of the work [12].

### 2.2. Characteristics of Ti Grade 2

Commercially pure titanium Grade 2, the chemical composition of which is given in Table 5, is characterized by relatively good formability, excellent corrosion resistance, and moderate strength.

The mechanical properties of Grade 2 titanium are as follows: YTS of 340 MPa, UTS of 430 MPa, and EL of 28% (Table 6). It should be added that titanium and its alloys exhibit a low modulus of elasticity that is slightly higher than that of aluminum and half that of steel and nickel alloys.

Grade 2 titanium has a relatively high melting point (below 1665 °C) and thermal conductivity lower than copper or aluminum (Table 7).

To ensure the integrity of titanium welds, the protective atmosphere should be maintained until the weld metal temperature drops below 426 °C. Before welding, it is essential that the weld joint surfaces be free of any contamination and that they remain clean during the entire welding operation. Contaminants such as oil, grease, and fingerprints must be eliminated with detergent cleaners or non-chlorinated solvents. Light surface oxides must be removed by acid pickling, while heavier oxides may require grit blasting, followed by pickling.

## 3. FEM Assumptions

A numerical model was built using the finite element method (FEM) and ADINA System v.9.8. Definitions of finite elements and the FEM problem-solving methodology used in this work are discussed in detail in [13,14,15]. To consider the physics of the RFW process, the influence of mechanical phenomena on thermal phenomena was included in the model. A thermomechanical, axial-symmetrical, and two-dimensional (2D) models of friction welding was developed. Due to axial symmetry, a 2D model is used to reduce the computational time. Figure 1 presents a schematic physical model of the analyzed issue. In this case, the aluminum part rotates and is pressed against the front side of the stationary titanium part. The rod has a diameter of 10 mm and a length of 20 mm. The left part is stationary, while the right part rotates around an axis. In the numerical model, the mesh was made using 4-node quadrangular elements with displacement and temperature coupling.

The ambient temperature T_A_ = 20 °C was used to determine the heat exchange between the component and environment and as the initial temperature of the components. It was also assumed that the welder receives heat from the components and that the constant ambient temperature there is at the element–machine interface. The physical model also assumes that heat is generated at the contact surface and that heat is exchanged between the components according to Equations (1) and (4) given in Section 3.2. As a result of the upsetting force, plastic deformation occurs, and heat is generated according to Equation (5). The adoption of an appropriate material model and a description of the behavior of the material in the range of temperatures occurring in the welding process is a very important issue. In this work, the elastic-plastic material model based on:The von Mises yield condition;An associated flow rule using the von Mises yield function;An isotropic, bilinear hardening rule,
was assumed in numerical modelling of the friction welding in the mechanical range.

### 3.1. Friction Conditions

The coefficient of friction μ is the key factor in developing a numerical model using FEM. Unfortunately, determination of the coefficient of friction both analytically and experimentally encounters great difficulties.

The first use of μ to represent the coefficient of friction has been attributed to a paper published in 1750 by Leonhard Euler (1707–1783) [16]. The author used variable material properties and a temperature-dependent coefficient of friction to calculate the temperature distribution and templates of the flash deformation in the friction welding of mild steel. Based on the literature review and our own research experience, in the FEM analysis for the Ti Grade 2/AA 5005 frictional pair, it was proposed to adopt the temperature-dependent coefficient of friction, as shown in Figure 2. The model predicts a constant coefficient of friction μ = 0.7 in the low temperature range (up to 100 °C) and close to zero μ = 0.01 above the melting point of AA 5005 (632 °C). In the case of dissimilar Ti Grade 2/AA 5005 materials, the smallest coefficient of friction was assumed for the material with lower melting point.

Between the extreme temperatures (100 °C and 632 °C), a linear dependence of the coefficient of friction was assumed. The model assumes that other factors, such as linear velocity or normal pressure, do not affect the value of the coefficient of friction. This approach seems to be the most beneficial due to the predictability of the heat flux, which depends to the greatest extent on changes in the coefficient of friction on the contact surface.

### 3.2. Thermal Conditions

The thermomechanical analysis took into account the frictional heat generated between the contact surfaces
q = F_f_·v,(1)
where q—frictional heat generated in time unit in the MES node; F_f_—frictional force; v—relative velocity of frictional surfaces.

The definition of frictional heat takes into account the energy dissipation and part of the heat transfer through each contact surface. The amount of heat that is transferred into contactor surface is f_c_·q, while the amount of heat that is transferred into the target surface is f_t_·q, where f_c_ and f_t_ are the parameters describing possible heat losses. The following relations exist between the parameters:0 ≤ f_t_ ≤ 1, 0 ≤ f_c_ ≤ 1, 0 ≤ f_t_ + f_c_ ≤ 1,(2)

In this work, f_c_ = 0.5 and f_t_ = 0.5 were assumed. The amount of heat generated depends on the coefficient of friction, the applied load, and the rotational speed of the components.

The boundary conditions of the thermal model are the conductivity between components, machine tooling, as well as the radiation and convection within the environment. The heat exchanged between the free surfaces of components and environment was described using the convection equation. The exchanged heat is described by the following equation:q^S^ = h_S_(T_A_ − T^S^),(3)
where h_S_—the coefficient of convection; T_A_—ambient temperature; T^S^—surface temperature of heat exchange.

The ambient temperature, and thus the initial temperature of the components, is determined by the temperature T_A_ = 20 °C. The coefficient of convective heat transfer between the welded front part and the environment was assumed to be at the level of h_S_ = 20 W/(m^2^·K), as in [17].

Heat exchanged between contact surfaces is described by an equation, which is similar to the equation describing convective heat transfer. The following equation describes heat introduced into the body with a lower temperature:q^L^ = h_c_(T^L^ − T^H^),(4)
where h_C_—heat transfer coefficient; T^L^ and T^S^—temperatures of the contact surfaces.

The internal heat Q, which is generated by plastic strain per unit time, is described as follows:(5)$$Q=\omega \sigma \;\dot{\varepsilon },$$
where ω—parameter describing the amount of plastic strain energy, which changes into heat 0 ≤ ω ≤ 1 (in this work, ω = 0.95 was assumed); σ—Cauchy stress; $\dot{\varepsilon }$—plastic strain rate.

### 3.3. Validation Issue

On the basis of the work [18] presenting a summary of approaches to the problem of FEM modeling, it was found that there is no uniform opinion on the validation of numerical calculations. The most frequent practice in this regard is to compare the measured temperature in the stationary component of the welded joint with numerical calculations. In this study, the measurement of the total length loss, also called the upset (l_u_), was used for validation. This approach requires only one parameter for analysis, which can be easily and accurately measured. In addition, the l_u_ value includes many factors affecting the process, in particular the dissimilar rheology and properties of the welded materials, as well as friction and thermal conditions. Before the model is validated, the heat transfer coefficient h_C_ must be determined. The accuracy of numerical calculations depends to the greatest extent on the adopted values describing the process. In dissimilar joints, due to the material or shape, the heat transfer coefficient h_C_ should not be neglected because the joint components dissipate heat from the friction area at different rates. Neglecting this factor for similar joints causes no major problems because the heat flux generated by friction is dissipated at the same rate through the components. In the case of dissimilar joints, a component with a lower value of thermal conductivity will dissipate heat more slowly from the friction zone. As a result, the temperature of this component will increase. The heat transfer coefficient h_C_ is responsible for equalizing the temperature between the components. A higher h_C_ value will result in less resistance to heat flow between components. Figure 3 shows the peak temperatures depending on the assumed h_C_ for the established numerical model. The values of peak temperatures decrease with the increase in the heat transfer coefficient h_C_. The relationship is non-linear and tends asymptotically to the value of 500 °C. From the value of h_C_ = 10^6^ W/(m^2^·K), the increase in peak temperatures is negligibly small. In this paper, it was assumed that there is no significant thermal barrier between the components, and the value of the heat transfer coefficient h_C_ = 10^6^ W/(m^2^·K) was adopted as a representative value.

The validation of numerical calculations was carried out together with the analysis of the sensitivity of the FEM mesh to the calculation results. The upset value of l_u_ = 2.3 mm obtained from the experiment was taken as the reference point. This value was 100% in relation to the upset value obtained from numerical calculations for the analyzed cases of the FEM mesh size. Figure 4 shows the relationship between the experimental results and the results of numerical calculations for the assumed size of FEM meshes.

The adopted mesh size resulted in the generation of a specific number of finite elements and nodes. Three FEM mesh sizes were used for the sensitivity analysis. The mesh sizes analyzed were 1 mm, 0.5 mm, and 0.1 mm. For the mesh size of 1 mm, 2 × 100 (200) axisymmetric solid elements were generated; for the mesh size of 0.5 mm, 2 × 400 (800) axisymmetric solid elements were generated; and for the mesh size of 0.1 mm, 2 × 10,000 (20,000) axisymmetric solid elements were generated. In the numerical analysis, 4-node quad linear elements were used. The 1 mm mesh generated 253 nodes, the 0.5 mm mesh generated 903 nodes, and the 0.1 mesh generated 20,503 nodes. Sensitivity analysis showed that decreasing the size of the FEM mesh leads to an increase in the computational value of upset l_uMES_. The mesh size below 0.5 mm was adopted in the calculations, as shown in Figure 5.

In the numerical analysis, a single time step of 0.01 s was assumed, which meant that the calculations for the 4 s rotary welding process required 400 time steps in both the thermal and mechanical modules.

### 3.4. Welding Parameters

The solid-state welding process of dissimilar materials should be carried out below the solidus melting temperature Ts of the material with the lower Ts value. In the case of the Ti Grade 2/AA 5005 joint, it is AA 5005 aluminum alloy with Ts = 632 °C. AA 5005 is sensitive to high temperatures ranging from 200 to 250 °C and may therefore lose some of its strength. In most cases, AA 5005 does not require hot working, but if it is required, the preferred temperature range is 204 ÷ 371 °C. The annealing temperature of AA 5005 aluminum alloy is 343 °C. Based on the thermal properties of AA 5005, it was assumed that a temperature T_0.6_ = 0.6Ts = ~380 °C at the faying surfaces allows for a good-quality weld to be obtained. In order to increase the safety margin, the RFW parameters at the temperature T_0.8_ = ~505 °C should be assumed for the faying surface.

#### 3.4.1. Friction Phase

In order to determine the basic parameters of the RFW process, i.e., the rotational speed n and the axial force F_A_ in the friction phase, the FEM numerical analysis covered the range of rotational speed n = 0 ÷ 3000 rpm and the axial force F_A_ = 0 ÷ 3000 N. It was assumed that in all analyzed cases, the time of the friction phase was t_fr_ = 1.5 s. The numerical calculation results were compiled in the form of a map of peak temperatures Tp on the faying surfaces as a function of the basic parameters (rotating speed n, rpm and axial forces F_A_, N) of the RFW process at the end of the friction phase, as shown in Figure 6. Temperature conditions are marked on the map with three colors. The area of peak temperatures Tp above the solidus melting point Ts is marked in red. In this case Tp > Ts (Ts = 632 °C for AA 5005). The application of process parameters from this area results in too high temperature at the faying surfaces. The area below Tp < T_0.6_ was marked in blue (T_0.6_ = 0.6Ts = ~380 °C). The application of process parameters from this area results in an insufficient temperature at the faying surfaces. In the remaining area marked in green, the process parameters allow a sufficient temperature to be obtained. In this area, the temperature line T_0.8_ = 0.8Ts = ~505 °C for AA 5005 was also marked.

In addition, Figure 6 shows areas of reduced quality of the welded structure resulting from poorly selected parameters of the RSW process. The hatched area on the left side of Figure 6 indicates a set of parameters for which the faying surfaces did not come into contact over the entire surface. As a consequence, welded joints with low mechanical strength are obtained. The hatched area on the right side of Figure 6 indicates a set of parameters for which sizable flash occurs due to a too large axial force relative to the material properties. This leads to a large total length loss.

#### 3.4.2. Forge Phase

The parameters analyzed by FEM in the forging phase F_A_ included the forging time t_f_ and the axial force. For the forging time t_f_, it was assumed that the peak temperature Tp on the faying surface should be lower than 200 °C. It was assumed that t_f_ > t_200_, where t_200_ is the time when the peak temperature Tp = 200 °C will be reached. Based on the literature review and our own experience, it was assumed that below 200 °C, diffusion processes between the joined surfaces basically do not occur.

The results of the initial FEM calculations are shown in Figure 7. The highest temperature in the forging phase occurred in the axis of the weldment on the faying surface of Ti Grade 2, at point P8, as marked in Figure 5. The dependence of temperature vs. time for point P8, in the entire cycle of the RFW process, allows for the precise determination of the peak temperature Tp = 200 °C and the corresponding time t_200_ = 1.7 s.

Finally, the value of t_f_ = 2.0 s was adopted as the forging time. Figure 7 shows the temperature distribution for the total welding time t_tw_ = 4.0 s, which corresponds to t_f_ = 2.0 s in the forging phase. The temperature distribution shows that the titanium rod dissipates heat more slowly than the aluminum one. Due to the difference in the thermal conductivity of the joined materials, there is a temperature discontinuity on the faying surfaces. Due to significant differences in the coefficient of thermal expansion, internal stresses will occur during cooling. For this reason, extension of the forging time above t_f_ > 2 s may be considered.

In order to determine the axial force in the forging phase, FEM calculations were performed for the range from F_A_ = 1000 N to F_A_ = 2000 N and forging time from t_f_ = 1 s to t_f_ = 4 s. In the analyzed range, it was assumed that the axial force increases linearly. At the points of the analyzed area, two requirements were checked after reaching the axial force F_A_ and the time t_f_. A decrease in the peak temperature below Tp < 200 °C was the first requirement, and an increase in the radius of the aluminum component not greater than r_1_ < 5.25 mm was the second requirement. In Figure 8, points that meet both requirements are marked in green, and points that do not meet these requirements are marked in red.

The key issue in this stage of the RFW process is to obtain the correct weldment without excessive flash. Appropriate selection of the axial force in the forging phase allows a high-strength weldment to be obtained. A linear increase in axial force over time is a better strategy than rapidly reaching the final value of the axial force and keeping it constant until the end of the forging time. The axial force should correlate with the temperature drop in the weldment.

When selecting the parameters of the RFW process, both in the analyzed case and in general, the parameters of the welding machine should be taken into account, especially the ability to control the process. For example, the application of high rotational speeds is generally associated with short times of the RFW process, which in practice may make it difficult to control the process. The previous analysis allowed for the selection of key parameters, such as rotational speed n, axial force F_A_ in the friction and forging phase, as well as friction times t_fr_ and forging t_f_. The key values of the process parameter as a function of the total welding time t_tw_ are presented in Figure 9.

The analysis of Figure 9 shows that the rotational speed from the beginning of the process to t_fr_ = 1.5 was n = 1440 rpm, after which the rotational speed was decelerated to zero within 0.5 s. In turn, the axial force increased for 0.5 s, from zero to the value F_A_ = 1000 N, which was kept at a constant level in the friction phase t_fr_ = 1.5 s as well as for 0.5 s until the complete stop of the rotary spindle.

As soon as the rotary spindle reached zero revolutions, the axial force in the forging phase began to increase to the value F_A_ = 1400 N for 2 s. The total time of the welding process was t_tw_ = 4.0 s.

## 4. Results

Most technological problems can be solved by numerical simulations of the process at the design stage. The analysis of the results of numerical calculations allows for the selection of almost all the parameters that have a decisive impact on the quality of the RFW process. This article presents the results of calculations showing how the basic parameters of the RFW process affect the temperature change, contact traction, and plastic strains in the weldment.

### 4.1. Temperature Changes in the Weldment

The temperature changes in the weldment for the key welding phases, which are shown in Figure 10, perfectly characterize the parameters of the RFW process. In addition to the temperature, the Figure 10 shows the axial shortening for the welding phases. The initial temperature of the components is shown for t_tw_ = 0 s, which is a reference for the axial shortening. For the time t_tw_ = 0.5 s, the temperature field in the weldment after reaching the constant axial force F_A_ = 1000 N in the friction phase is shown. The axial shortening at that time was marked as l_05_ and was only l_05_ = 0.114 mm. Despite the short welding time, a significant increase in the peak temperature Tp = 314.9 °C was noted in relation to the initial temperature of the components. It is clearly visible that the largest area with temperatures above the initial temperature is in the aluminum component. The end of the friction phase occurs at t_tw_ = 1.5 s, and it means the beginning of the braking of the rotating spindle. In this phase, constant revolutions (n = 1440 rpm) generate the most frictional heat, which is dissipated to the components. Therefore, the temperature peak is the highest and is Tp = 513.5 °C. Changes in mechanical properties occurring under the influence of temperature and the action of a constant axial force F_A_ = 1000 N cause plastic deformation in the weldment.

In this phase of the process, it is clearly visible how the flash is formed in the aluminum component. Intensive flash formation results in a large value of the burn-off length l_bo_ = 1.761 mm. At the time t_tw_ = 2.0, the spindle of the friction welding machine came to a complete stop. This is also the beginning of the forging phase in which the axial force F_A_ increases. At this time, a decrease in the peak temperature Tp = 390.9 °C is observed due to the lack of frictional heat. The increase in axial shortening at this time is small, l_20_ = 1.944 mm. The end of the RFW process at t_tw_ = 4.0 s is characterized by a decrease in the peak temperature to Tp = 180.5 °C. Although the increasing axial force F_A_ = 1400 N causes plastic deformations in the weldment, the increase in temperature caused by plastic strains is imperceptible. There is also no significant increase in the flash, and the axial shortening is minimal in relation to the values observed at t_tw_ = 2.0. The axial shortening for the entire RFW process, called the upset, was l_u_ = 2.009 mm.

#### 4.1.1. Temperature Changes at the Points

The analysis of temperature changes over time at the characteristic points shown in Figure 5 gives even more complete information about the nature of the temperature changes in the weldment. Figure 11, shows the temperature–time profiles for points P1–P5 of the AA 5005 component. The temperatures at points P1–P3 do not have the same values for the same times. For the time t_tw_ = 0.5 s, the temperature difference between P1 and P2 reaches ΔT = ~100 °C. Since the points lie on the same contact surface, this will mean uneven heat generation.

The highest peak temperature in the AA 5005 component Tp = 484 °C was recorded for point P2, located in the middle of the rod radius, which means that the most heat was generated around this point. Point P3, located in the axis of the rod on the contact surface, had the lowest peak temperature of Tp = 447 °C among the three points located on the contact surface. Point P1, located on the edge of the contact surface, heats up slower than points P2 and P3. In the temperature profile for this point, around time t_tw_ = 0.6 s, a local decrease in temperature associated with flash formation is observed.

After this time, point P1 is outside the rubbing surface and heats up only through conduction in the aluminum rod. The peak temperatures recorded at points P4 and P5, located in the axis of the AA 5005 rod and 1 mm and 2 mm from the contact surface, respectively, differ from each other by ΔTp = ~20 °C. A higher peak temperature of Tp = 440 °C is observed at point P4.

Figure 12 shows the temperature–time profiles for points P5–P10 of the Ti Grade 2 component. The analysis of the temperature profile for points P6-P8 shows that for the time t_tw_ ≈ 0.5 s, the temperature difference between P6 and P7 is ΔT = ~200 °C and is twice as large as in the case of the aluminum rod. The highest peak temperature in the Ti Grade 2 component Tp = 514 °C was recorded for point P6, located in the middle of the rod radius, which confirms previous observations that the most heat was generated around this point. However, the temperature profile shows that until the time t_tw_ = 0.5 s was reached, point P6 heated up slower than points P7, P8, and even P9.

Point P8, located in the axis of the rod on the contact surface, had the lowest peak temperature Tp = 447 °C among the three points located on the contact surface. In the initial phase of friction, point P6, located on the edge of the contact surface, heats up slower than points P7 and P8, but after t_tw_ = 0.5 s, the temperature increase is very intense until the highest peak temperature of all the analyzed points is reached. Point P6, after t_tw_ = 0.5 s, is still in the contact zone, and since the Ti Grade 2 component deforms to a small extent and dissipates heat less, the temperature at this point increases significantly. The peak temperatures recorded at points P9 and P10, lying in the axis of the Ti Grade 2 rod and 1 mm and 2 mm from the contact surface, respectively, differ significantly by ΔT = ~70 °C. The higher peak temperature of Tp = 372 °C is observed for point P9. The difference in temperature values compared to the AA 5005 rod is significant due to the difference in thermal conductivity between the materials.

#### 4.1.2. Temperature Changes at the Faying Surface

The relationship between the temperature and the change in the shape of the L1 contact surface of the AA 5005 component in the Y direction is shown in Figure 13. Additionally, the graph shows the peak temperature migration at the specific time points for this component.

In turn, Figure 14 shows the relationship between the temperature and the change in the shape of the contact surface L2 of the Ti Grade 2 component in the Y direction. In addition, the graph shows the migration of the peak temperature at the specified time points for this component.

As can be seen from the presented graphs, the peak temperatures are not always the same on the surfaces in contact during welding. In the friction phase, a flash is formed, and the front surface of the AA 5005 rod increases. Despite this, the temperature peaks on both surfaces are located at a similar distance from the center of the rod and have similar values. It is only in the forging phase, when the weldment cools down, that the peak temperatures on the contact surfaces differ. In the Ti Grade 2 rod, the temperature peaks are located in the rod axis, while in the AA 5005 rod, they migrate to about 2 mm from the rod axis.

### 4.2. Contact Traction Changes on the Faying Surfaces

Changes occurring on the contact surface can be characterized by analyzing the distribution of contact traction in the individual phases of the RFW process. Figure 15 shows the distribution of contact traction for respective times t_tw_ on the faying surfaces of the welded components.

As results from the analysis of the contact traction distribution up to the time t_tw_ = 0.5 s, there are areas on the contact surfaces with a contact traction value of zero. After this time, the contact traction distribution becomes more homogeneous, and non-zero values occur on the entirety of the faying surfaces.

In the friction phase, a shift in the peak of the contact traction from the center of the rod to its edge is observed. The highest peak of the contact traction, in the friction phase, occurs for the time t_tw_ = 0.4 s and is σ_Max_ = 143 MPa. Its front face moves and increases its surface, creating the flash. The greatest deformations related to the change in the external shape of the weldment occur in the friction phase.

A characteristic feature of the distribution of the contact traction after the time t_tw_ = 0.5 s is the increasing peak of the contact traction in the area of the edge of the Ti Grade 2 component. In the forging phase, starting from time t_tw_ = 2 s, the temperature in the weldment decreases and begins to stabilize in the volume of the weldment.

As a result of cooling, the lateral layers of the weldment become more rigid than the rod axis and transfer most of the loads close to the edge of the component. Due to these factors, an increase in the contact traction peak is observed near the edge with increasing axial force. As can be seen at the end of the forging phase, the maximum contact traction peak is σ_Max_ = 213 MPa.

### 4.3. Weldment Deformation

The distribution of plastic strains, ε, which is the result of the axial force and the changes in material properties under the influence of temperature, illustrates the deformation of the weldment well. Significant plastic strains occur in the AA 5005 component, while the Ti Grade 2 component is not subject to plastic strains. The distribution of plastic strains at characteristic points of the total weld time is shown in Figure 16. As can be seen, up to t_tw_ = 0.5 s, the increase in plastic strains is small, and the peak of plastic strain does not exceed ε_Max_ < 0.06. The highest intensity of changes in plastic deformation occurs in the friction phase between the time t_tw_ = 0.5 s and t_tw_ = 1.5 s. During this one-second period, the axial force has a constant value F_A_ = 1000 N and the rotational speed maintains a constant value n = 1440 rpm. The peak of plastic deformation at the end of this period is ε_Max_ = 0.4992.

The half-second period between the friction and forging phase from t_tw_ = 1.5 s to t_tw_ = 2 s is characterized by a decrease in rotational speed to zero at the constant axial force. In this period, the increase in plastic strains is small, and at the end of this period, it is ε_Max_ = 0.5074. The last phase of the RFW process, i.e., the forging phase, is characterized by an increase in axial force without the spindle rotating. In this phase, the plastic strain’s increase is small. With the cooling of the weldment, the peak of plastic strains of the AA 5005 component moves from the center of the rod to the edge of the Ti Grade 2 component and at the end of the forging process is ε_Max_ = 0.5088.

## 5. Discussion

As shown so far in rotary friction welding (RFW), the weldment temperature can change depending on the dissimilarity of the components being welded. The amount of heat generated during the welding process is a result of the frictional energy generated between the two rotating parts. When welding dissimilar materials, the coefficient of friction, which is a measure of the sliding resistance, changes over the contact surface. This affects the amount of heat generated during the welding process and the resulting weldment temperature. Additionally, the thermal conductivity and heat capacity of the two materials can also influence the temperature distribution within the weldment. In general, the local temperature at the interface of the two materials can reach high values, while the temperature of the bulk material remains relatively low. The temperature profile across the interface depends on the welding parameters, the thermal properties of the materials, and the time–temperature history. It should be noted that the heat-affected zone (HAZ) in an RFW is typically small, which means that the thermal impact on the base material (BM) is limited.

### Effect of Dissimilarity in Material Properties on RFW Process Parameters

The analysis of the results in Figure 11 and Figure 14 shows that the process parameters should be selected regarding the material that first undergoes plastic deformation under the influence of temperature. In the analyzed case, the AA 5005 aluminum alloy has a lower melting point and a higher thermal conductivity coefficient. This combination of material properties results in faster heat dissipation from the friction zone by the AA 5005 component. As shown in Figure 17, the component made of Ti Grade 2 will dissipate heat more slowly, so the temperature at point P6 will be higher in the friction phase than at point P1, just as at the end of the forging phase, the temperature at point P9 will also be higher than at P4.

The slower heating rate of P6 point at the beginning of the process results from the lack of contact between the faying surfaces and, therefore, the lack of frictional heat. After t_tw_ = 0.5 s the temperature increases rapidly because the faying surfaces come into full contact, allowing frictional heat to be generated in the areas farthest from the axis of the rod.

#### Temperature Peak Migration

The peak temperature on the faying surface migrates during the RFW process. As shown in Figure 13, the migration of the peak temperature in the friction phase, on the surface of L1 AA 5005 occurs from the half of the rod radius to the outside. However, in the forging phase, in the absence of a source of frictional heat, the temperature peak returns to the inside of the rod. In the case of Ti Grade 2 rod, the temperature peak has a higher value than for AA 5005 material. This is due to uneven heat dissipation from the weldment.

The higher temperature in the friction phase at the maximum radius of the rod results from the high linear velocity causing the release of more heat than at smaller radii. The temperature difference between the highest and lowest value on the L1 surface at t_tw_ = 1.5 s reaches ΔT = 65 °C. For the L2 surface, there is also a temperature difference and for the same time, which is ΔT = 68 °C. The slight difference between the ΔT values for the L1 and L2 surfaces results from the formation of flash and the difference in thermal conductivity of the welded materials.

The formation of a flash is associated with the occurrence of increasing friction forces. This leads to the heating of the contact layers and the transport of the material outwards from the axis of the rod. The rotation of the aluminum rod causes the oxides to migrate. Layers of unoxidized base materials are continuously exposed as the aluminum flash grows. Aluminum and titanium oxides are removed from the welded surfaces as the aluminum flows out. This explains the tendency of the new chemically active layers of aluminum and titanium to form an adhesive and diffusion joint. Alumina is removed together with the flash material, while rutile is removed under the influence of tangential forces because there is no plastic deformation of the surface to which it adheres. The oxide-free surfaces are the source of a strong metallic bond. Metallographic tests were carried out to determine the presence of oxides on the surface and to determine the presence of IMCs. No harmful compounds were found in the tests. In Figure 18, the microstructure of the Ti Grade 2/AA 5005 junction etched with 1% HF can be observed.

Determining the safe level of plastic deformation and material flow, as well as determining the safe level of heat generation inhomogeneity along the radial direction, promotes a lowered occurrence of harmful compounds and a greater uniformity of the mechanical properties of the joint. IMC formation at the interface was dominated by welding parameters, including friction load, burn length, and upsetting load.

In order to obtain the correct joint, the tests focused on the analysis of joint strength when welding parameters were changed. Guided by the above guidelines, experimental analysis, and numerical analysis, it was assumed that the optimal parameters for the weldment of Ti Grade 2 and AA 5005 occur under the assumption that the temperature peak generated by the heat of friction on the contact surfaces will be:T_0.8_ = 0.8Ts = ~505 °C,(6)
where Ts is the solidus temperature for AA 5005.

Another parameter defined for the RFW process is the maximum axial force occurring in the forging phase, determined on the basis of the yield strength of the AA 5005 material at the temperature T_0.8_ = 505 °C.

Numerical analysis showed that the adoption of the axial force is determined according to the relationship:F_AMax_ = Re(T_0.8_)πr^2^ = 1400 N,(7)
where Re(T_0.8_)—yield point AA 5005 at temperature T_0.8_ = 505 °C; πr^2^—face area of the rod with radius r.

In the case of the friction phase, the axial force F_A_ was determined in the form of the following relationship:F_A_ = 0.7F_AMax_ = 980 N,(8)

In this paper, a slightly higher value of the axial force in the friction phase F_A_ = 1000 N was assumed. In turn, the rotational speed n = 1440 rpm was determined on the basis of Figure 6 using the T0.8 isothermal line.

## 6. Conclusions

The paper provides an insight into the individual effects of volumetric changes resulting from the change in parameters of the RFW process. Based on the numerical analyses and experimental tests performed for the rotational speed n = 0–3000 rpm and axial force F_A_ = 0–3000 N, the dependencies needed for the correct selection of parameters in the joining process of Ti Grade 2 and AA 5005 were determined. The following crucial conclusions can be highlighted:Numerical calculations strongly depend on the assumed frictional conditions and the heat transfer coefficient h_C_. Good results were obtained assuming a low coefficient of friction at the solidus temperature of the material with a lower melting point. The optimal value of the heat transfer coefficient was h_C_ = 10^6^ W/m^2^·K.On the faying surfaces, there is an uneven temperature distribution during the RFW process predicted by numerical simulations.The peak temperature on the faying surfaces migrates during the RFW process. In the friction phase on the faying surfaces, it moves from the half of the rod radius to the outside, and in the forging phase, in the absence of a source of frictional heat, the temperature peak returns to the inside of the rod.Good-quality Ti Grade 2/AA 5005 joints, without IMCs, were obtained with the assumption that the temperature peak at the end of the friction phase will be around 80% of the solidus temperature of the AA 5005 material.The maximum axial force F_AMax_ in the forging phase should be the product of the yield strength at temperature T_0.8_ and the face area of the rod.The axial force F_A_ in the friction phase should be about 70% of the F_Amax_.

## Figures and Tables

**Figure 1 materials-16-04787-f001:**
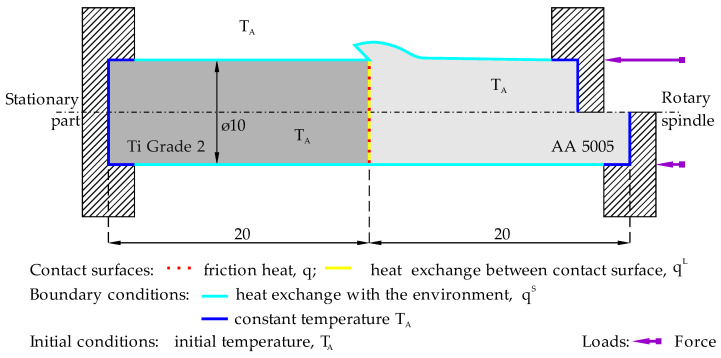
Diagram of the physical model of the RFW process.

**Figure 2 materials-16-04787-f002:**
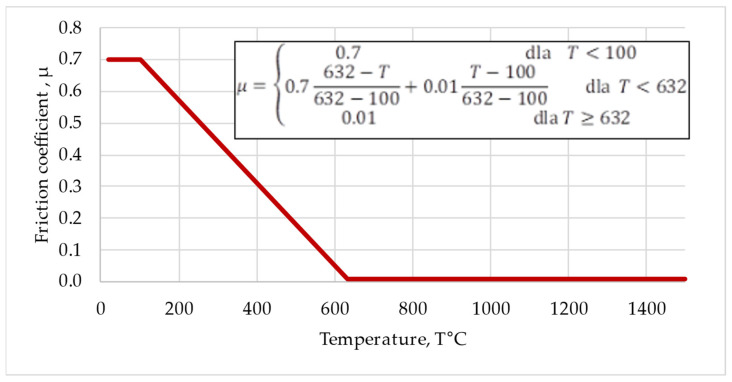
Coefficient of friction vs. temperature adopted for friction pair of Ti Grade 2/AA 5005.

**Figure 3 materials-16-04787-f003:**
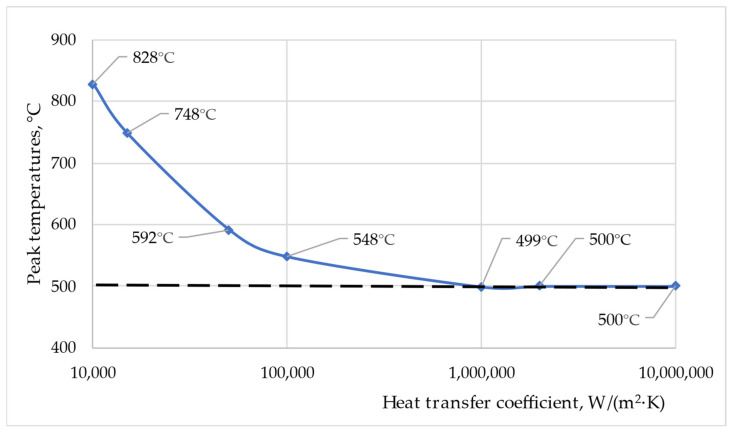
Peak temperatures, °C, depending on assumed h_C_, W/(m^2^·K).

**Figure 4 materials-16-04787-f004:**
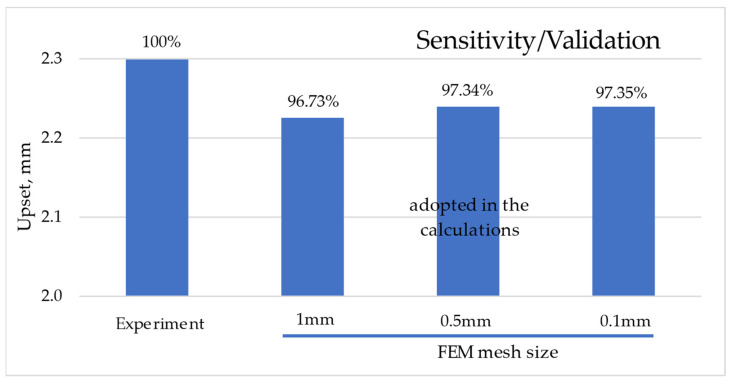
Sensitivity of the FEM mesh size to the upset value.

**Figure 5 materials-16-04787-f005:**
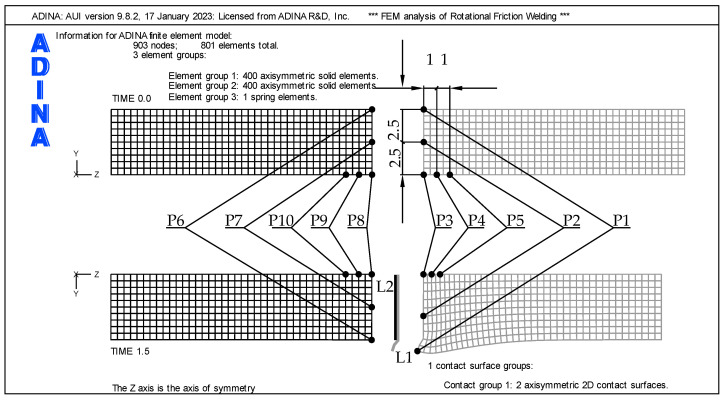
Information for ADINA finite element model.

**Figure 6 materials-16-04787-f006:**
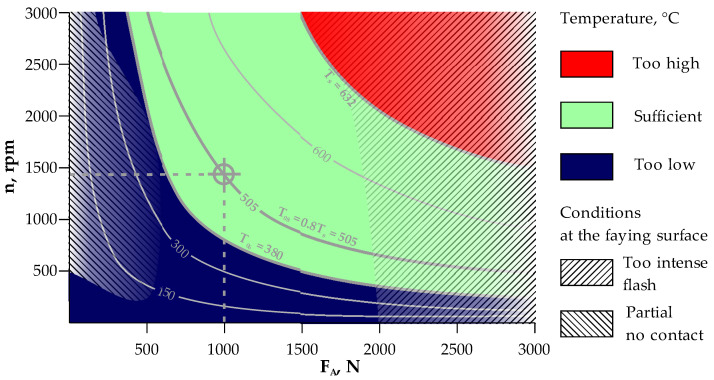
Peak temperatures, T_p_, at the faying surfaces as a function of the basic parameters of the RFW process at the end of the friction phase.

**Figure 7 materials-16-04787-f007:**
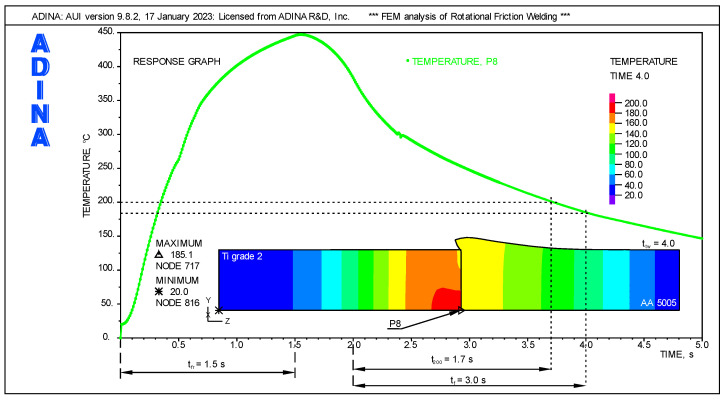
Determination of the minimum forging time t_f_.

**Figure 8 materials-16-04787-f008:**
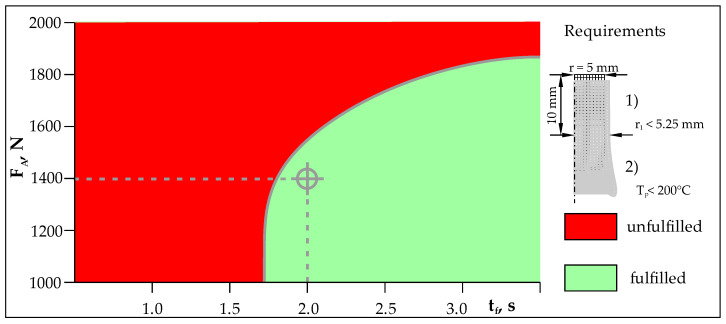
Range of analyzed RFW parameters in the forging phase.

**Figure 9 materials-16-04787-f009:**
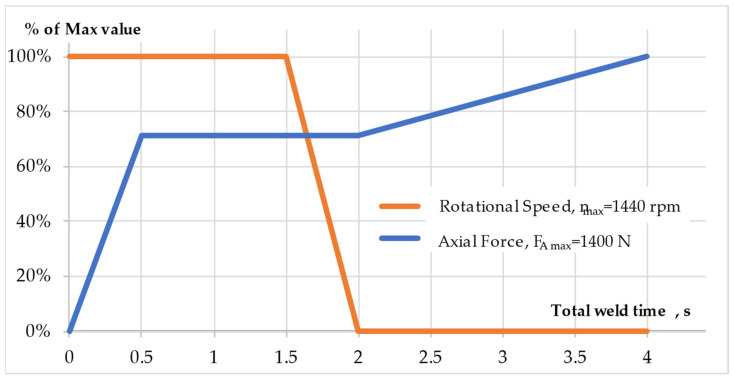
Key values of RFW process parameters.

**Figure 10 materials-16-04787-f010:**
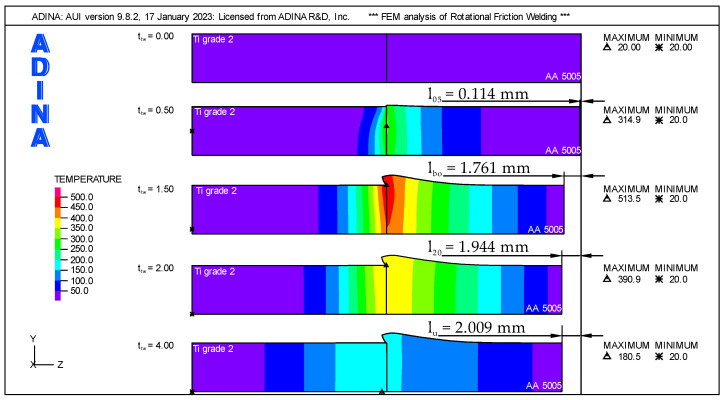
Temperature distributions and axial shortening in weldment for key welding phases.

**Figure 11 materials-16-04787-f011:**
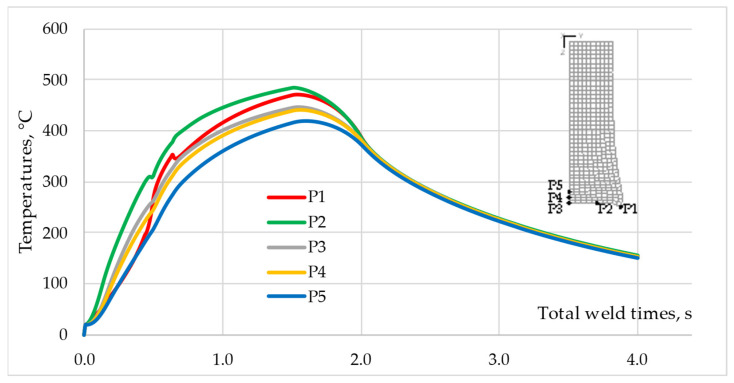
Temperature-time profile for points P1–P5 of the AA 5005 component.

**Figure 12 materials-16-04787-f012:**
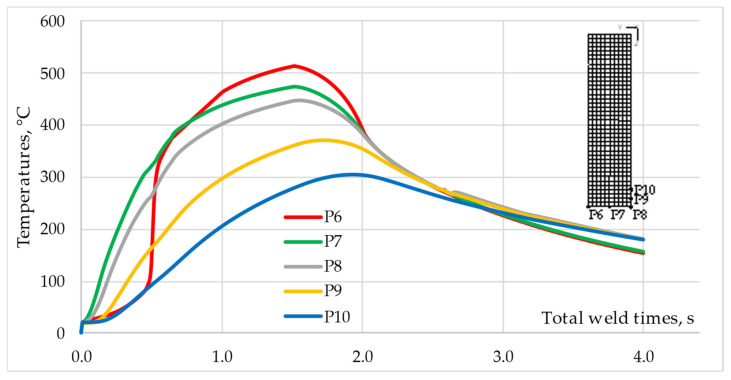
Temperature–time profiles for points P6–P10 of Ti Grade 2 component.

**Figure 13 materials-16-04787-f013:**
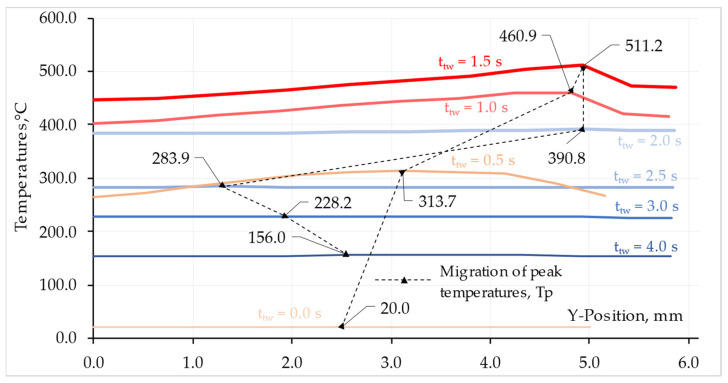
Peak temperature, Tp, migration on L1 AA 5005 surface.

**Figure 14 materials-16-04787-f014:**
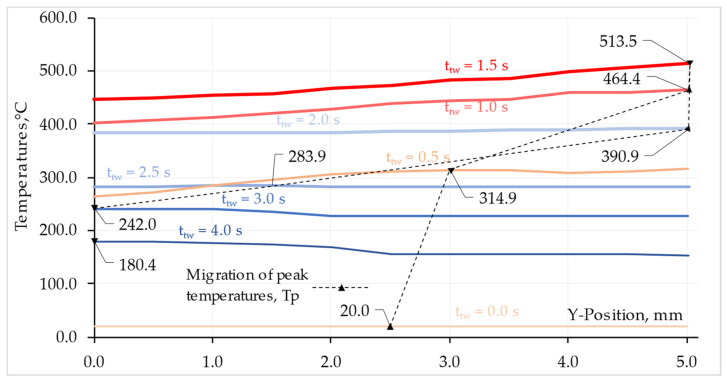
Peak temperature, Tp, migration on L2 Ti Grade 2 surface.

**Figure 15 materials-16-04787-f015:**
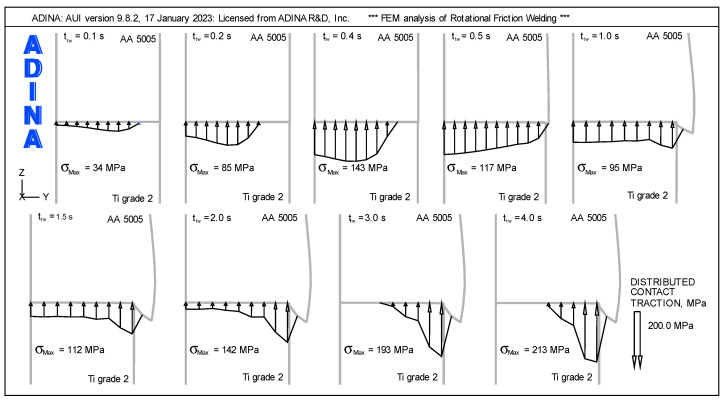
Distribution of contact traction for selected times t_tw_ on the faying surfaces.

**Figure 16 materials-16-04787-f016:**
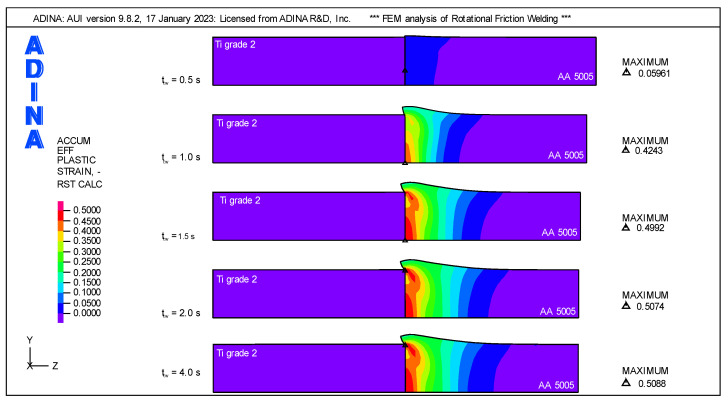
Distribution of plastic strains in weldment for key welding phases.

**Figure 17 materials-16-04787-f017:**
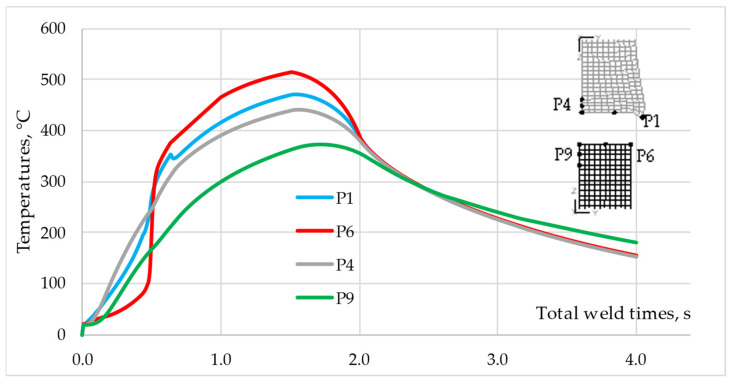
Comparison of the temperature profiles for points P1 and P6.

**Figure 18 materials-16-04787-f018:**
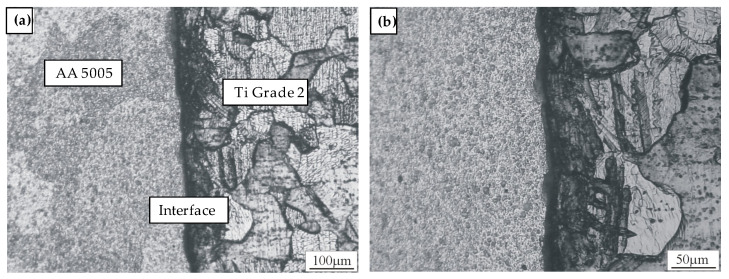
Microstructure of the Ti Grade 2/AA 5005 junction; (**a**) free of IMCs interface; (**b**) etched with 1% HF.

**Table 1 materials-16-04787-t001:** Exemplary conditions of RFW process for various combinations of aluminum and titanium alloys.

Ref	Rotating Component ^1^	Stationary Component ^1^	Speed	Load ^2^		Time or the Burn-Off Length ^3^	
				Friction Phase	Forging Phase	Friction Phase	Forging Phase
[5]	Al_2_O_3_; rod; ϕ10 mm	AA 6061; rod ϕ10 mm	14,500 rpm	18 MPa	46 MPa	0.85 s	3.5 s
[6]	Al_2_O_3_; rod; ϕ10 mm	Aluminum; rod; ϕ10 mm	14,000 rpm	18 MPa	46 MPa	1.4 s	1.6 s
[7]	AA 6061; rod; ϕ12.5 mm	AISI 1018; rod; ϕ12.5 mm	4200 rpm	23 MPa	50–60 MPa	1 s	5 s
[8]	AA 2024; rod; ϕ12 mm	AA 2024; rod; ϕ12 mm	2200 rpm	40 MPa	90–110 MPa	4–8 s	-
[9]	Ti Grade 2; tube; ϕ60 mm; 3.9 mm	Ti Grade 2; tube; ϕ60 mm; 3.9 mm	2200 rpm	20 kN	30 kN	32 s	22 s
[10,11]	Ti Grade 23; rod; ϕ6 mm	Ti Grade 23; rod; ϕ6 mm	2500 rpm	4 kN	6 kN	1.5 mm	9 s
[1]	Ti Grade 5; rod; ϕ8 mm	AA 6082; rod; ϕ8 mm	6000 rpm–14,000 rpm	9 kN–18 kN	13 kN–30 kN	Max 3 s	0.25 s–2.5 s
[2,3]	Ti- α; rod; ϕ10 mm	AA 5005; rod; ϕ10 mm	1440 rpm	1 kN–10 kN	25 kN–35 kN	2 s–5 s	0.5 s–2.5 s

^1^ The following elements are: material; shape of component; outer diameter; tube wall thickness. ^2^ The load can be expressed in MPa or kN depending on the welding machine control system used. ^3^ Time is expressed in s and burn-off length in mm.

**Table 2 materials-16-04787-t002:** Chemical composition of AA 5005-H32 determined by optical emission spectrometer.

Material	Element Content, wt.%							
	Al	Cr	Cu	Fe	Mg	Mn	Si	Zn
AA 5005-H32	Bal.	0.09	0.19	0.67	1.07	0.18	0.27	0.23

**Table 3 materials-16-04787-t003:** Mechanical properties of AA 5005-H32.

Material	Density, kg/m^3^	YTS, MPa	UTS, MPa	EL, %	Young’s Modulus, GPa	Poisson’s Ratio, -	Shear Modulus, GPa	Shear Strength, MPa	Hardness, Brinell
AA 5005-H32	2700	117	138	11	68.9	0.33	25.9	96.5	36

**Table 4 materials-16-04787-t004:** Thermal properties of AA 5005-H32.

Material	Thermal Conductivity, W/mK	Specific Heat Capacity, J/KgK	Melting Point, °C		CTE Linear, 10^−6^/°C
			Solidus	Liquidus	
AA 5005-H32	200 *	900	632	654	21.9

* at room temperature.

**Table 5 materials-16-04787-t005:** Chemical compositions of Ti Grade 2 according to ASTM B 265 specification for titanium and titanium alloy strip, sheet, and plate.

Material	Element Content, wt.%					
	Ti	C	H	Fe	N	O
Ti Grade 2	Bal.	0.06	0.01	0.20	0.03	0.12

**Table 6 materials-16-04787-t006:** Mechanical properties of CP Ti Grade 2 in annealed condition.

Material	Density, kg/m^3^	YTS, MPa	UTS, MPa	EaB, %	Young’s Modulus, GPa	Poisson’s Ratio, -	Shear Modulus, GPa	Shear Strength, MPa	Hardness, Brinell
Ti Grade 2	4510	340	430	28	102	0.34	38.0	380	200

**Table 7 materials-16-04787-t007:** Thermal properties of Ti Grade 2.

Material	Thermal Conductivity, W/mK	Specific Heat Capacity, J/KgK	Transformation Temperature, °C		CTE Linear, 10^−6^/°C
			Beta Transus	Liquidus	
Ti Grade 2	16.4 *	523	913	1665	8.6

* at room temperature.

## Data Availability

The data presented in this study are available on request from the corresponding author.

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
