# Peer review of "A New Method of Predicting the Parameters of the Rotational Friction Welding Process Based on the Determination of the Frictional Heat Transfer in Ti Grade 2/AA 5005 Joints"

_materials, 2023, doi:10.3390/ma16134787_

Round 1

Reviewer 1 Report

This paper reported to analyze the variation of temperature, stress and joint formation in rotational friction welded Al alloy/Ti alloy joint via numerical simulation, from which the optimization of welding process could be proposed in order to further enhance the corresponding properties of the dissimilar metals joints. I think it is an interesting paper that could provide some insights for the research community. However, I do have some comments on this paper:

1.       In the introduction part, the author has already offered a relatively complete literature review for RFW. I cannot find a reason why the author put some paragraphs on FSW. Should be deleted.

2.       I don’t think it is quite right to make the comment that a higher temperature at the weld interface can be achieved at higher rotational speed by citing these two papers. As one cited paper made some comments on the other cited paper that it actually did not propose such argument. It may be true that before reaching the equilibrium condition, a higher speed could result in a higher heating rate, thus a higher temperature, but after the equilibrium condition (probably takes a long time), the link between temperature and speed would become insignificant. This may be more obvious at a high rotation speed as a less time would take to reach such condition. Authors may should refer to this article https://doi.org/10.1016/j.jmapro.2019.08.001 for more discussion.

3.       Some mistakes in using symbols are spotted. Please check the whole submission.

4.       Similarly, the statement that “the maximum interface temperature decreases slightly with the increase of axial pressure.” is not correct. How to define slightly? As far as I am concerned, by altering the pressure, the temperature would show quite substantial change, especially at the heating stage. So the difference should be specified.

5.       The surface modification: sometimes the existence of texture would enlarge the contact area, however, sometimes it would create the non-bonded region. These references may be useful: https://doi.org/10.1016/j.matchar.2021.111347

6.       For the comparison of post-processing, these references could be added: https://doi.org/10.1016/j.jmatprotec.2015.08.004

7.       I think the CDFW is better than DDFW as it is more commonly used in the literature.

8.       I don’t quite understand the temperature map in Fig. 11. It is from simulation results, how come the temperature could go beyond 632 which is the melt temperature of Al alloy as the friction coefficient is approaching 0 considering the friction time is rather short (1.5s). More clarification should be made.

No issues have been identified.

Author Response

I am submitting a revised manuscript for consideration for publication in Materials. The manuscript entitled “A new method of predicting the parameters of the Rotational Friction Welding process based on the determination of the frictional heat transfer in the Ti Grade 2 /AA 5005 joints” has been thoroughly reviewed considering reviewer comments. In particular, the number of pages of the introduction and references has been reduced.

I would like to thank the reviewers for their time and valuable comments, which have certainly improved this publication.

#Reviever 1

This paper reported to analyze the variation of temperature, stress and joint formation in rotational friction welded Al alloy/Ti alloy joint via numerical simulation, from which the optimization of welding process could be proposed in order to further enhance the corresponding properties of the dissimilar metals joints. I think it is an interesting paper that could provide some insights for the research community. However, I do have some comments on this paper:

  1. In the introduction part, the author has already offered a relatively complete literature review for RFW. I cannot find a reason why the author put some paragraphs on FSW. Should be deleted.

As suggested by the reviewer, the paragraphs concerning FSW were removed from the introduction

  1. I don’t think it is quite right to make the comment that a higher temperature at the weld interface can be achieved at higher rotational speed by citing these two papers. As one cited paper made some comments on the other cited paper that it actually did not propose such argument. It may be true that before reaching the equilibrium condition, a higher speed could result in a higher heating rate, thus a higher temperature, but after the equilibrium condition (probably takes a long time), the link between temperature and speed would become insignificant. This may be more obvious at a high rotation speed as a less time would take to reach such condition. Authors may should refer to this article https://doi.org/10.1016/j.jmapro.2019.08.001  for more discussion.

From the context of the question, I'm guessing that the question is about this particular part of the publication's text:

The process duration is inversely proportional to the rotational speed, while the in-tensity of heat generation is directly proportional to rotational speed. In the works [15,25] it was revealed that higher temperature at the weld interface can be achieved at higher rotational speed, which results in earlier material extrusion, and thus increases the rate of upsetting.

I fully agree with the reviewer's comment. I do not refer to this text in the work, because as part of the reduction of the introduction, this fragment of the text and the cited literature were removed. However, the literature for more discussion indicated in the note is cited in the work. In the context of the coefficient of friction, it occurs on line 256.

  1. Some mistakes in using symbols are spotted. Please check the whole submission.

The subsection on terms and symbols was removed as part of the introduction reduction and at the request of the 2nd reviewer. The correctness of symbols in the remaining text was checked.

  1. Similarly, the statement that “the maximum interface temperature decreases slightly with the increase of axial pressure.” is not correct. How to define slightly? As far as I am concerned, by altering the pressure, the temperature would show quite substantial change, especially at the heating stage. So the difference should be specified.

The moot piece of text has been removed as part of the reduction of the introduction.

  1. The surface modification: sometimes the existence of texture would enlarge the contact area, however, sometimes it would create the non-bonded region. These references may be useful: https://doi.org/10.1016/j.matchar.2021. 111347 

Updated the paragraph with the following text:

surface modification: according to [10], increasing the contact between the surfaces of the parts to be welded by increasing their roughness, e.g. by means of texturing or laser peening improve the bonding strength of the weld. sometimes the existence of texture would enlarge the contact area, however, sometimes it would create the non-bonded region [15].

  1. For the comparison of post-processing, these references could be added: https://doi.org/10.1016/j.jmatprotec.2015.08.004 

The paragraph on post-weld heat treatment has been supplemented with recommended literature.

  1. I think the CDFW is better than DDFW as it is more commonly used in the literature.

 Comparative considerations regarding the difference between CDFW and DDFW has been removed as part of the reduction of the introduction.

  1. I don’t quite understand the temperature map in Fig. 11. It is from simulation results, how come the temperature could go beyond 632 which is the melt temperature of Al alloy as the friction coefficient is approaching 0 considering the friction time is rather short (1.5s). More clarification should be made.

In order to determine the basic parameters of the RFW process i.e., the rotational speed n and the axial force FA in the friction phase, the FEM numerical analysis covered the range of rotational speed from n = 500 rpm to n = 3000 rpm every 500 rpm and the axial force FA = 500 N  FA = 3000 N every 500 N. This gives a total of 36 cases of FEM calculations. It was assumed that in all analyzed cases the time of the friction phase was tfr = 1.5 s. The numerical calculation results were compiled in the form of a map of peak temperatures Tp on the faying surfaces as a function of the basic parameters (rotating speed n, rpm and axial forces FA, N) of the RFW process at the end of the friction phase, as shown in Figure 11. The points of peak temperature values listed on the map were approximated by a smooth surface. It should be noted that the values marked in red (above Ts=632°C) do not result directly from numerical calculations but are only the result of approximation by a smooth surface.

Reviewer 2 Report

1. The manuscript contains 34 pages. This is a lot for an article describing the results of a not very complex simulation.

2. The bibliographic list contains 83 sources. A significant part of the literature is weakly related to the simulation of friction welding of aluminum and titanium.

3. Rotational welding has been known for more than 60 years, so it is not worth explaining the meanings of various terms and ISO process parameters.

4. Literature review alone took 8 pages.

5. The authors do not explain the shortcomings of the previously conducted modeling options by different authors and the difference between their methodology and those previously used.

6. The first simulation results appear only on page 17, this can be allowed in a monograph or in a textbook, but not in a scientific article.

7. Some drawings are difficult to interpret. For example, the authors have shown in Fig. 12 6 visualization color scales for 3 figures without explaining what the visualization means. Part of the data is common to the three figures shown (speed of rotation, welding time), so they can be placed in the caption. Instead, the authors repeat them three times in the figure field.

8. The authors do not explain why in the proposed model the area with the maximum temperature is located on the axis (figure 13), that is, in the zone of zero rotation speeds.

9. I can't understand figure 21. Contact stress diagrams are shown on it. The authors used the color of the arrows and their different lengths. The authors do not explain the purpose of using two imaging options at the same time. The color scale has a maximum value of 90 MPa, although the authors indicate maximum values up to 213 MPa.

10. The authors present the distribution of effective plastic deformation on the figure. 22, although the radial and axial deformations separately are much more interesting for understanding the welding process.

11. The analysis of determining the possibilities for the formation of intermetallic compounds in the welding zone was carried out very superficially.

Author Response

I am submitting a revised manuscript for consideration for publication in Materials. The manuscript entitled “A new method of predicting the parameters of the Rotational Friction Welding process based on the determination of the frictional heat transfer in the Ti Grade 2 /AA 5005 joints” has been thoroughly reviewed considering reviewer comments. In particular, the number of pages of the introduction and references has been reduced.

I would like to thank the reviewers for their time and valuable comments, which have certainly improved this publication.

# Reviever 2

  1. The manuscript contains 34 pages. This is a lot for an article describing the results of a not very complex simulation.

The manuscript has been redrafting, and 4 pages of text were removed.

  1. The bibliographic list contains 83 sources. A significant part of the literature is weakly related to the simulation of friction welding of aluminum and titanium.

Due to the redrafting of the text, the number of literature items has decreased to 54 items.

  1. Rotational welding has been known for more than 60 years, so it is not worth explaining the meanings of various terms and ISO process parameters.

Due to the rewording of the text, explanations of the meaning of various terms and parameters of the ISO process have been removed.

  1. Literature review alone took 8 pages.

Due to the redrafting of the text, the literature review was reduced to 3 pages of text.

  1. The authors do not explain the shortcomings of the previously conducted modeling options by different authors and the difference between their methodology and those previously used.

Please note that the description of the numerical model takes up a significant part of the work, 10 pages not counting the description of the material model and material data. References to modeling options by different authors are cited in more than 25 references ([5]; [6]; [8]; [25]; [27]; [33-35] [36]; [37]; [38]; [40]; [41-44]; [45-47]; [48]; [49]; [50]; [51]; [52]; [53]).

 The most important aspects of modeling are distinguished by subchapters:

3.1 Friction conditions

The chapter describes the author's approach to the modeling of friction conditions.

3.2. Thermal conditions

Where it was proposed to include heat exchanged between contact surfaces because in the analyzed literature, the hc coefficient was usually omitted, as in the work [49] or its value is not given.

3.3. Validation issue

This section shows that in dissimilar joints, due to the material or shape, the heat transfer coefficient hc should not be neglected, because the joint components dissipate heat from the friction area at different rates.

  1. The first simulation results appear only on page 17, this can be allowed in a monograph or in a textbook, but not in a scientific article.

The manuscript has been redrafting, and simulation results appear earlier.

  1. Some drawings are difficult to interpret. For example, the authors have shown in Fig. 12 6 visualization color scales for 3 figures without explaining what the visualization means. Part of the data is common to the three figures shown (speed of rotation, welding time), so they can be placed in the caption. Instead, the authors repeat them three times in the figure field.

Removed color scale for contact tractions values.

Due to the large dispersion of values, different scales of plastic deformation values were used for individual drawings.

The caption under the drawing was changed to:

Impact of the axial force on the quality of the weldment at a constant rotational speed n=1400rpm and friction time tfr=1.5s.

  1. The authors do not explain why in the proposed model the area with the maximum temperature is located on the axis (figure 13), that is, in the zone of zero rotation speeds.

The temperature map shown in the figure shows the temperature distribution for the time ttw=4.0, i.e. 3.5 s after the end of friction. During this time, the temperature is equalized due to conduction. However, in the case of a titanium rod, it occurs more slowly due to the much lower thermal conductivity of titanium in relation to aluminum. For comparison, please see Figure 15.

  1. I can't understand figure 21. Contact stress diagrams are shown on it. The authors used the color of the arrows and their different lengths. The authors do not explain the purpose of using two imaging options at the same time. The color scale has a maximum value of 90 MPa, although the authors indicate maximum values up to 213 MPa.

Values above 200 MPa occur at the edge of the titanium rod. Narrowing the scale down to 90 MPa is to better illustrate lower values in the friction phase.

  1. The authors present the distribution of effective plastic deformation on the figure. 22, although the radial and axial deformations separately are much more interesting for understanding the welding process.

In authors opinion the distribution of plastic strains ε, which is the result of the axial force and changes in material properties under the influence of temperature, illustrates well the deformation of the weldment. However, considering the reviewer's comment in future papers, the authors will make every effort to show the radial and axial deformations separately.

  1. The analysis of determining the possibilities for the formation of intermetallic compounds in the welding zone was carried out very superficially.

Controlling the formation of brittle IMC's at the joint interface seems to be key to achieving good quality joints. Their presence and growth at the joining surfaces significantly reduces the load-bearing capacity. Of the intermetallic phases from the Ti-Al binary system (Ti3Al, TiAl, Al3Ti, Al2Ti, and Al5Ti), the Al3Ti phase is usually expected in dissimilar titanium/aluminum joints. It results from its thermodynamic stability related to the diffusion of aluminum in titanium as a dominant process. The development of Al3Ti phase is considered to be a cause of joint cracking near the interface of Al and Ti - for example, in the case of FSW process.

Round 2

Reviewer 2 Report

1. The manuscript contains 28 pages. This is a lot for an article describing the results of a not very complex simulation.

2. The bibliographic list contains 54 sources. A significant part of the literature is weakly related to the simulation of friction welding of aluminum and titanium. For example, articles [22] - [24] are devoted to welding aluminum with steel. The probability of formation of intermetallic compounds in this pair of welded materials is much higher due to the absence of a strong oxide film on steel, which is characteristic of titanium alloys.

3. The authors do not explain the shortcomings of the previously conducted modeling options by different authors. Hence, the purpose of this article is not clearly formulated. The first indications of the shortcomings of the previously used models appear on page 11.

4. Part of the data that should be in the literature review turned out to be under simulation conditions. For example, a description of the different ways to set the friction coefficient takes up a page (3.1. Friction conditions). As a result, a rather strange friction coefficient of 0.7 was chosen at temperatures up to 100 °C.

5. The authors use the Log true strain label below the graphs in figures 3 and 4. Did they really mean the logarithm? But then for log = 0 true strain = 1. This clearly contradicts the graphs shown.

6. It is not clear what the authors wanted to say in paragraph 3.4. Welding parameters. All changes to the properties of the AA 5005 aluminum alloy have already been set based on figure 3.

7. Some drawings are difficult to interpret. For example, figure 11 contains a large amount of unnecessary data: the background of the program window itself, the pressure in the friction zone, the rotation speed and friction time repeated three times. Moreover, the rotation speed and time are given in the caption. The authors showed the gap between AA5005 and Ti grade 2 at an axial force of 500 N, but did not explain its origin.

8. The authors do not explain why in the proposed model the area with the maximum temperature is located on the axis (figure 12), where rotation speeds is zero. In reality, at the beginning of the growth of axial force (total weld time 2 s), the maximum temperature is observed at points P1 and P2 (figure 16).

9. I can't understand figure 21. Contact stress diagrams are shown on it. The authors used the color of the arrows and their different lengths. The authors do not explain the purpose of using two imaging options at the same time. The color scale has a maximum value of 90 MPa, although the authors indicate maximum values up to 213 MPa.

10. The authors present the distribution of effective plastic deformation on the figure. 22, although the radial and axial deformations separately are much more interesting for understanding the welding process.

11. The analysis of determining the possibilities for the formation of intermetallic compounds in the welding zone was carried out very superficially. They did not analyze in any way the works related to the latent period of the formation of intermetallic compounds at various temperatures. In addition, the presence of discrete strong fragments of intermetallic compounds in a ductile aluminum matrix can improve the properties of the near-boundary layer.

Author Response

I am submitting a revised manuscript for consideration for publication in Materials. The manuscript entitled “A new method of predicting the parameters of the Rotational Friction Welding process based on the determination of the frictional heat transfer in the Ti Grade 2 /AA 5005 joints” has been thoroughly reviewed considering reviewer comments.

In the re-edited document, the literature and the number of pages were significantly reduced at the expense of discussion threads raised by the reviewer.

I would like to thank the reviewers for their time and valuable comments, which have certainly improved this publication.

# Reviever 2

  1. The manuscript contains 28 pages. This is a lot for an article describing the results of a not very complex simulation.

Reduced the number of document pages to 20.

  1. The bibliographic list contains 54 sources. A significant part of the literature is weakly related to the simulation of friction welding of aluminum and titanium. For example, articles [22] - [24] are devoted to welding aluminum with steel. The probability of formation of intermetallic compounds in this pair of welded materials is much higher due to the absence of a strong oxide film on steel, which is characteristic of titanium alloys.

The number of document references has been reduced to 18 items.

  1. The authors do not explain the shortcomings of the previously conducted modeling options by different authors. Hence, the purpose of this article is not clearly formulated. The first indications of the shortcomings of the previously used models appear on page 11.

The most important aspects of modeling are distinguished by subchapters:

3.1 Friction conditions

The chapter describes the author's approach to the modeling of friction conditions.

3.2. Thermal conditions

Where it was proposed to include heat exchanged between contact surfaces because in the analyzed literature, the hc coefficient was usually omitted, as in the work [49] or its value is not given.

3.3. Validation issue

This section shows that in dissimilar joints, due to the material or shape, the heat transfer coefficient hc should not be neglected, because the joint components dissipate heat from the friction area at different rates.

  1. Part of the data that should be in the literature review turned out to be under simulation conditions. For example, a description of the different ways to set the friction coefficient takes up a page (3.1. Friction conditions). As a result, a rather strange friction coefficient of 0.7 was chosen at temperatures up to 100 °C.

Repetitive information in the text of the document has been reduced.

  1. The authors use the Log true strain label below the graphs in figures 3 and 4. Did they really mean the logarithm? But then for log = 0 true strain = 1. This clearly contradicts the graphs shown.

Due to the redaction of the text, the controversial figures have been removed, only the reference to the literature item has been left.

  1. It is not clear what the authors wanted to say in paragraph 3.4. Welding parameters. All changes to the properties of the AA 5005 aluminum alloy have already been set based on figure 3.

Due to the redaction of the text, the controversial figures have been removed.

  1. Some drawings are difficult to interpret. For example, figure 11 contains a large amount of unnecessary data: the background of the program window itself, the pressure in the friction zone, the rotation speed and friction time repeated three times. Moreover, the rotation speed and time are given in the caption. The authors showed the gap between AA5005 and Ti grade 2 at an axial force of 500 N, but did not explain its origin.

Due to the redaction of the text, the controversial figures have been removed.

  1. The authors do not explain why in the proposed model the area with the maximum temperature is located on the axis (figure 12), where rotation speeds is zero. In reality, at the beginning of the growth of axial force (total weld time 2 s), the maximum temperature is observed at points P1 and P2 (figure 16).

The temperature map shown in the figure 7 (current numbering) shows the temperature distribution for the time ttw=4.0, i.e. 3.5 s after the end of friction. During this time, the temperature is equalized due to conduction. However, in the case of a titanium rod, it occurs more slowly due to the much lower thermal conductivity of titanium in relation to aluminum. For comparison, please see Figure 10.

  1. I can't understand figure 21. Contact stress diagrams are shown on it. The authors used the color of the arrows and their different lengths. The authors do not explain the purpose of using two imaging options at the same time. The color scale has a maximum value of 90 MPa, although the authors indicate maximum values up to 213 MPa.

Due to the redaction of the text, the controversial figures have been removed.

  1. The authors present the distribution of effective plastic deformation on the figure. 22, although the radial and axial deformations separately are much more interesting for understanding the welding process.

In authors opinion the distribution of plastic strains ε, which is the result of the axial force and changes in material properties under the influence of temperature, illustrates well the deformation of the weldment. However, considering the reviewer's comment in future papers, the authors will make every effort to show the radial and axial deformations separately.

  1. The analysis of determining the possibilities for the formation of intermetallic compounds in the welding zone was carried out very superficially. They did not analyze in any way the works related to the latent period of the formation of intermetallic compounds at various temperatures. In addition, the presence of discrete strong fragments of intermetallic compounds in a ductile aluminum matrix can improve the properties of the near-boundary layer.

Due to the redrafting of the text, the discussion thread on intermetallic phases was removed, leaving only the conclusion that no harmful compounds were found in the tests.